# Sexual Dysfunction and Atopic Dermatitis: A Systematic Review

**DOI:** 10.3390/life11121314

**Published:** 2021-11-29

**Authors:** Laura Linares-Gonzalez, Ignacio Lozano-Lozano, Luis Gutierrez-Rojas, Mario Lozano-Lozano, Teresa Rodenas-Herranz, Ricardo Ruiz-Villaverde

**Affiliations:** 1Department of Dermatology, Hospital Universitario San Cecilio, 18016 Granada, Spain; ilozano@ugr.es (I.L.-L.); teresa.rodenas.sspa@juntadeandalucia.es (T.R.-H.); ricardo.ruiz.villaverde.sspa@juntadeandalucia.es (R.R.-V.); 2Biohealth Research Institute in Granada (ibs.GRANADA), 18012 Granada, Spain; mlozano@ugr.es; 3Department of Psychiatry, University of Granada, 18011 Granada, Spain; gutierrezrojas@ugr.es; 4Psychiatry Service, Hospital Clínico San Cecilio, 18016 Granada, Spain; 5CTS-549 Research Group, Institute of Neuroscience, University of Granada, 18016 Granada, Spain; 6Department of Physical Therapy, Faculty of Health Sciences, University of Granada, 18011 Granada, Spain; 7Sport and Health Joint University Institute (iMUDS), University of Granada, 18011 Granada, Spain; 8‘Cuídate’ Support Unit for Oncology Patients, 18011 Granada, Spain

**Keywords:** atopic dermatitis, sexual dysfunction

## Abstract

Atopic dermatitis (AD) is a chronic inflammatory disease of the skin whose main symptom is pruritus and may affect all age ranges. Regarding the prevalence, it has been estimated at around 10% of the world population. Many concomitant diseases have been associated with AD, but the causal relationship between AD and psychological impairment has not been clearly established. Scientific literature studying the probable association between male or female sexual dysfunction and dermatological pathology is limited, even more so in AD. This systematic review was conducted following the Preferred Reporting Items for Systematic Reviews and Meta-Analyses (PRISMA) reporting guidelines and the Cochrane Collaboration methodology for systematic reviews. All relevant articles in English were identified through a search from inception to 10 December 2020, including the following databases: Medline (via PubMed), Scopus, Web of Science Core Collection, and SciELO. The results of the search were compiled using the COVIDENCE software for systematic reviews. The methodological quality of the included studies was done using the “Quality Assessment Tool for Observational Cohort and Cross-Sectional Studies” and the “Quality Assessment of Case-Control Studies” developed by the National Heart, Lung, and Blood Institute, National Institutes of Health (NIH). Our search yielded potentially relevant studies. Five studies that evaluated the prevalence of sexual dysfunction in atopic dermatitis were retrieved after applying the selection criteria. The present systematic review achieved data from 8088 patients with atopic dermatitis from four articles. Sample sizes for atopic dermatitis patients ranged from 266 to 3997. We identified one cohort study with four years of follow-up, three studies with a cross-sectional design, and one case-control study. Three studies reported data disaggregated by the severity of atopic dermatitis. Two studies included healthy controls with a total sample size of 1,747,755 subjects. Two studies compared data with other dermatological conditions such as psoriasis. In conclusion, we can establish that unlike other psychological comorbidities such as anxiety and depression, sexual dysfunction is a field scarcely explored in the literature. This sexual dysfunction focuses on the male sex in large population studies and in clinical diagnoses without exploring it through specific and validated questionnaires in this regard. Further studies focused on both genders are needed. It is important to correlate this sexual dysfunction with the severity of the disease, previous treatments, and cardiovascular comorbidities.

## 1. Introduction

Atopic dermatitis (AD) is a chronic inflammatory disease of the skin whose main symptom is pruritus and may affect all age ranges. The form of presentation differs depending on this, being the most frequent sign of eczema [1].

Regarding the prevalence, it has been estimated at around 10% of the world population. It also varies according to age group and gender. AD is more common in children (around 10–20%), with males prevailing during this stage, reducing the prevalence rate to 1–5% in adulthood (when it is more frequent around 30 years), where the female sex predominates. In this last age group, this pathology is more severe [1,2], defining it by the use of immunosuppressants or biological agents as the target treatment or by the need for hospitalization due to an outbreak of AD [3,4]. Given the high prevalence, it is important to carry out a rigorous study of this condition, considering the wide variety of comorbidities and their impact on the quality of life of patients who suffer from it.

The clinical presentation of AD in the adult population can vary depending on different criteria such as age, ethnicity, and the underlying biological mechanisms [5].

The disease has three suitably differentiated forms of presentation: (a) a persistent form that begins in childhood and has a chronic-recurrent course until adulthood; (b) a relapsing form, which begins its symptoms in childhood and improves but regresses after a few years free from disease, and (c) a form that begins in adulthood. In a study recently published by Nettis et al., AD appeared for the first time in childhood in 63.3% (if we group both the persistent form and the recurrent form), while 36.7% showed its appearance directly in adulthood [5].

There are different treatments available to control AD, including moisturizers, topical steroids, topical immunomodulators (pimecrolimus, tacrolimus), phototherapy, and systemic immunomodulators when topical therapy or the extension of the disease make their assistance necessary (cyclosporin, methotrexate, mycophenolate mofetil). It is not uncommon for patients to be dissatisfied with their treatment, especially in the moderate and severe forms and precisely in these forms, the portfolio of available drugs will be expanded in the coming years with the inclusion of anti-JAK (Baricitinib, Upadacitinib, and Abrocitinib) and anti-IL drugs (Lebrikizumab, Nemolizumab, Risankizumab, and tralokinumab) apart from already marketed Dupilumab [6].

AD represents a significant physical and psychological burden for both patients and their families. The impact of the disease on the quality of life of patients with AD could be related to its severity [7]. Many of the patients with AD report that the disease limits their daily life, as well as lead them to reduce and avoid social contacts and numerous activities. The feeling of dissatisfaction with life is present in 17% of people with AD, increasing in those who suffer from the most severe form [8]. In the latter, more prevalently, we observed a greater loss of quality-adjusted life years [9]. That is why interventions at the pharmacological and psychotherapeutic levels have favorable results in improving the quality of life of these patients in the treatment of AD and concomitant diseases.

Many concomitant diseases have been associated with AD [3,4,5,6,7,8,9], but the causal relationship between AD and psychological impairment has not been clearly established. It is difficult to decide whether psychological conditions are a direct product of AD or different disorders. Despite this, a relationship between AD with psychological disorders such as anxiety, depression, or even a high risk of suicide has been widely explored [10]. It is important to highlight that not only can AD play a fundamental role in these types of pathologies, but there is a certain relationship between them and atopic diseases in general. A correlation has been observed between atopic pathologies with anxiety and depression in the family context so that the diagnosis of the atopic disease in one of the siblings of the same family should lead us to study the psychological comorbidity both in the diagnosed individual and the whole family [11]. Similarly, not only a qualitative association between AD and psychological disorders has been observed, but the more severe the skin pathology is, the greater the emotional burden may also be [12]. In this way, a medical action that adds a psychological study of these patients is important, as it has been observed that the improvement of AD symptoms leads to a reduction of the emotional burden [13].

When evaluating the concomitance of psychological disorders, the association of AD with sexual dysfunction should also be studied. Male sexual dysfunction is mainly represented by erectile dysfunction (ED). The European Association of Urology defines ED as “the persistent inability to achieve and maintain an erection sufficient to allow satisfactory sexual performance” [14]. The prevalence of ED is high throughout the world, reaching nearly 50% in advanced adulthood, and shares certain risk factors with diseases of a cardiovascular nature. Therefore, we cannot forget that erection is a process in which both psychological and neurovascular phenomena converge when studying a possible association with AD since it can present comorbidities in both aspects. We must bear in mind that, like AD, ED has an important impact on the quality of life of the patient and family [13]. There are numerous ways to measure ED. Without a doubt, questionnaires are the most practical methods for evaluating this condition in daily clinical practice. One of the most widely used is the IIEF-5 (International Index Erectile Function), which classifies ED as mild, mild to moderate, moderate, and severe.

Regarding female sexual dysfunction, we should consider the six domains of the female sexual sphere: desire, arousal, lubrication, orgasm, pain, and satisfaction [14]. The DSM-5 establishes a series of criteria to define female sexual dysfunction, based on the previous six domains, defining it as “a lack or reduction of sexual interest or excitement” and this can be manifested by an absence or reduction of interest in sexual activity, decrease in erotic thoughts, reduction of sexual pleasure with sexual activity, or absence or decrease of sensations during sexual encounters [15]. The prevalence is also high, reaching over 40% and becoming higher in postmenopausal women, the most frequent condition in both European and North American women being related to alterations in the sphere of desire, while the prevalence related to alterations in the orgasm dominance drops to around 20–30% [16]. Equally to ED, we have a wide variety of methods to measure female sexual dysfunction and, in the same way as in male pathology, questionnaires are the best option, so it can be evaluated through the Sexual Function Index Feminine created by Rosen and collaborators, based on the six domains previously mentioned.

Both female and male sexual dysfunction represent a symptom and not a disease. Additionally, in both population groups, we must bear in mind that sexuality represents an externalization of well-being on a physical, social, and psychological level. It is, therefore, necessary to rigorously evaluate patients to study possible underlying pathologies and disorders that could be the origin of both ED and female sexual dysfunction [13,14].

Scientific literature studying the probable association between male or female sexual dysfunction and dermatological pathology is limited, even more so in AD. That is why studies on the subject in question are scarce [17]. The role of systemic inflammation derived from AD concerning neurovascular alteration that prevents erection in ED has also been studied [18]. Similarly, the role of testosterone and its low levels in AD patients seems to establish a causal link with ED [19]. Other paths lead to a possible alteration in the psychological sphere without being mutually exclusive. It is, therefore, imperative that clinical staff attend to complaints related to the sexual sphere that come from patients with AD or other skin disorders. Therefore, the objective of this systematic review is to synthesize the prevalence of sexual dysfunction in atopic dermatitis and to know which are the most used instruments of evaluation of sexual dysfunction in this population.

## 2. Materials and Methods

### 2.1. Data Sources and Search Strategy

This systematic review was conducted following the Preferred Reporting Items for Systematic Reviews and Meta-Analyses (PRISMA) [20] reporting guidelines and the Cochrane Collaboration methodology for systematic reviews [21]. All relevant articles in the English language were identified through a search from inception to 10 December 2020, including the following databases: Medline (via PubMed), Scopus, Web of Science Core Collection, and SciELO; in consultation with a research librarian. A manual search of all bibliographies of the included articles was also carried out. To allow the systematic review to identify all studies evaluating the association between the presence of sexual disorder in atopic dermatitis, we did not limit the initial search to any type of study. The search was divided into two main sections: to locate studies on atopic dermatitis, a Boolean search was performed using the “dermatitis, atopic” MeSH headings combined with free entry terms with the Boolean OR operator; to identify sexual dysfunction, a Boolean search was performed using the following MeSH headings combined with the Boolean OR operator: “sexual dysfunctions, psychological”, “sexual dysfunction, physiological”, “Dyspareunia”, “Erectile Dysfunction”, “Premature Ejaculation”, “Sexual and Gender Disorders”, “Vaginismus”, “impotence, vasculogenic”. To ensure the identification of all available papers, synonyms were also used as free entry terms and combined with Boolean operator OR. Both topics were combined with the Boolean operator AND to answer the research question posed in this systematic review.

### 2.2. Study Selection and Data Extraction

The results of the search were compiled using the COVIDENCE software for systematic reviews [22], through which the entire process of selection and identification of articles to be included was carried out. After removing duplicates, two independent researchers (MLL and RRV) conducted the initial screening by title and abstract and the second screening by full text, and differences were resolved via discussion (only one article needed a third researcher (ILL). Studies were included if they met all of the following inclusion/exclusion criteria: (1) prospective, retrospective, and cross-sectional studies evaluating the prevalence of any sexual disorder in atopic dermatitis; (2) original data from an original study; (3) no time limit; (4) published in any language; (5) sexual dysfunction evaluated with a validated scale or diagnosed established by ICD-8–10 CM (International Classification of diseases, 8, 9, 10th Revision, Clinical modification). Studies that were only published as abstracts were excluded.

A data extraction form was developed for the extraction of the key elements of each study. The following variables were extracted from each full-text article: year and country of publication, study design, sample size, severity of atopic dermatitis, presence of control group, method of evaluation of sexual dysfunction, and major study findings.

### 2.3. Quality Assessment: Risk of Bias Evaluation

The methodological quality of the included studies was done using the “Quality Assessment Tool for Observational Cohort and Cross-Sectional Studies” and the “Quality Assessment of Case-Control Studies” developed by the National Heart, Lung, and Blood Institute, National Institutes of Health (NIH) (available from https://www.nhlbi.nih.gov/health-topics/study-quality-assessment-tools, accessed on 24 November 2021). The tool is designed to aid the appraisal of internal validity of cross-sectional, cohort studies, and case-control studies. It comprises 14 criteria (12 for the case-control version). All criteria can be answered as “yes”, “no”, “cannot determine”, “not applicable”, or “not reported”. All responses other than “yes” indicate a risk of bias. Inherent to the design, cross-sectional studies automatically score “not applicable” on criteria 6, 7, 10, and 13. Two reviewers (MLL and RRV) carried out the evaluation of the methodological quality of the included studies [23]. Any disagreement among the reviewers was resolved by meeting and discussing among all the authors to establish a consensus.

Due to the heterogeneity of the instruments used, the methodological differences of the studies included, and the number of studies, the quantitative combination of results was not possible. Similarly, because there were fewer than 9–10 studies per specific outcome within each questionnaire type, publication bias assessment was not possible.

## 3. Results

Our search yielded potentially relevant studies. Five studies that evaluated the prevalence of sexual dysfunction in atopic dermatitis were retrieved after applying the selection criteria [17,18,24,25,26]. The complete information about study selection and reasons for exclusion are available in Figure 1. Three studies were removed because sufficient data were not available, and attempts to contact the corresponding authors were not successful.

The details and characteristics of these studies are included in Table 1. The majority of studies (*n* = 4) were from Europe [17,24,25,26], and only one was from Asia. The present systematic review achieved data from 8106 patients with atopic dermatitis from five articles. Sample sizes for atopic dermatitis patients ranged from 266 [24] to 3997 [18]. We identified one cohort study with four years of follow-up [17], three studies with a cross-sectional design [24,25,26], and one case-control study [18]. Three studies reported data disaggregated by the severity of atopic dermatitis [17,24]. Two studies included healthy controls with a total sample size of 1,747,755 subjects [17,18]. Two studies compared data with other dermatological conditions such as psoriasis [17,25].

Two of the studies only refer to the male population when studying erectile dysfunction in patients with AD [17,18]. In these studies, the mean age was 53.0 and 50.6 years. Most studies found a prevalence of sexual dysfunction in atopic dermatitis greater than 10%. The prevalence of sexual dysfunction ranged from 6.7% reported at baseline in the Egeberg et al. study [17] to 57.9% reported by Misery et al. [26]. In the study led by Sampogna et al. [25], the highest prevalence of sexual dysfunction was established between 40 and 59 years of age. The prevalence of sexual dysfunction in atopic dermatitis was higher than that reported for healthy subjects in the studies that included them (below 6.7%), although it is lower compared to other pathologies such as psoriasis (above 34%). Misery et al. [25] concluded that patients with genital involvement reported higher disease burden and lower QoL, being associated with severe AD.

The instruments used to assess sexual dysfunction were very variable. The most commonly used sexual dysfunction assessment questionnaire was the Dermatological Life Quality Index, which includes one question about this disorder (question nine) but is not a validated scale for sexual dysfunction. One of the studies uses one validated questionnaire to determine sexual dysfunction (CHOQ) added to DLQI [24], and two only rely on a question from a questionnaire (DLQI) [25,26]. A study establishes the presence of sexual dysfunction based on the treatment prescription used for this pathology [17]. In all the studies, sexual dysfunction was considered in patients over 18 years of age [17,18,24,25,26].

Evaluation of the risk of bias of the studies can be found in Figure 2. In our evaluation, all the studies included were of good (*n* = 2) or fair (*n* = 3) methodological quality. Specific limitations included the lack of justification of the previous sample size, heterogeneity in the recruitment of the control population, the cross-sectional design of four of the studies evaluated, or the heterogeneity in the measurement instrument used to evaluate the outcome.

## 4. Discussion

Conforming to the literature, more than 30% of patients with AD have significant psychiatric and psychosocial comorbidities [27]. An increased risk of depression and anxiety [11,28,29,30], risk of suicide [31], and sleep disorders [32,33] have been studied and tested using validated questionnaires. However, sexual problem assessment and sexual dysfunction (SD) are usually only insufficiently recorded. Sexual health can markedly affect the quality of life and interpersonal relationships, but it is frequently ignored and undervalued, even more in women [33,34].

Ludwig et al. [35] have recently published a review on the relationship of AD with sexual health. It establishes how AD affects sexual health, quality of life, male and female sexual dysfunction, effects on sexual partners, and how it impacts physical intimacy. We have focused our review on sexual dysfunction itself and how to measure it, which we will discuss below, showing the great variability of concept and measure existing today. In light of the results of this systematic review, this lack of knowledge about sexual dysfunction in AD seems to be evident. Furthermore, despite the small amount of scientific information on the subject, our results highlight the importance of this comorbidity in AD.

A variety of medical, psychological, and lifestyle factors have been implicated in the etiology of SD. The need for validated measures to diagnose and quantify a complex entity such as SD has led to the generation of numerous scales and self-report questionnaires, some of which focus specifically on the patient’s or partner’s sexual function, whereas others focus on more subjective aspects of well-being or distress related to sexuality [33,34]. A major challenge in the assessment of SD is for the clinician or researcher to select the appropriate questionnaire among a large number of measures available [35]. In this context, our systematic review revealed a serious lack of standardized, internationally acceptable questionnaires that are thoroughly epidemiologically validated in reasonably large, randomized, representative samples and that could be used to assess SD in people with or without a partner and independently of the partner’s gender [36]. These measurements provide useful, quantitative information but don’t cover all aspects of sexual function as currently defined. Another concern of relevance in sexual medicine is the potential to raise socially desirable responses to certain questions [36,37] as well as the discomfort that a physician or patient may feel [34].

This systematic review examined the impact of AD on sexual function. The study by Misery et al. [24] suggested that AD considerably affects sexual desire and that the impact of AD is significant for both the patient and the partner. This impact is clearly greater in patients with severe AD and is correlated with those patients who score above 8 on the DLQI. This finding has been endorsed years later in a larger cohort where it relates the severity of AD with the clinical involvement of the genital area and sexual dysfunction [26]. Similarly, one study by Long et al. [38] observed that 19% of AD subjects reported that their sex lives were affected by the disease. Ebata et al., after quantifying serum levels of sexual hormones, concluded that the prevalence of hypogonadism was significantly higher in 40 patients diagnosed with AD than that in age-matched healthy controls. This was evidenced as patients had lower serum levels of testosterone, free testosterone, and higher levels of the luteinizing hormone than controls [39]. Egeberg et al. [18] concluded that there was an independent association between erectile dysfunction (ED) and prior AD based on a case-control study by comparing the risk of prior AD between patients with newly diagnosed ED and matched controls. The systemic inflammation experienced by many of these patients may contribute to endothelial dysfunction, which is central to the pathophysiology of ED. However, Egeberg et al. [17] in 2017 performed a cross-sectional study to estimate the prevalence and odds ratio of erectile dysfunction in patients with AD and found that the prevalence of erectile dysfunction in men with AD was significantly lower, whereas their risk of new-onset erectile dysfunction was not increased compared with the general male population. Unlike the previously mentioned cross-sectional studies, Sampogna established that sexual difficulties are related to young age, high severity, anxiety, and depression [25]. Regarding women, we were not able to identify similar studies with which to compare the results of our patients. To our knowledge, the present study was the first to focus on the sexual functioning of patients with AD of both sexes specifically.

Atopic dermatitis (AD) is just one of the many chronic dermatoses in which there is a significant impact on the sexual life of patients who suffer from it as well as their partners [33]. In the literature, there are studies reporting the effect of psoriasis on sexual function since 1988 [40]. Many studies have associated psoriasis with a significantly increased prevalence and risk of new-onset dysfunction erectile [40,41,42,43,44,45,46]. Of note, psoriasis can alter patients’ body image, resulting in low self-esteem and stigmatization, leading in turn to SD [37]. Similarly, the sexual function seems to be altered in women with psoriasis, although studies are much scarcer in this area [46]. Hidradenitis suppurativa is another inflammatory dermatosis that can be attached to SD. The prevalence of SD in HS was reported in several studies. Overall, compared with healthy controls, patients with HS reported more sexual function problems that contribute to a lower quality of life [47,48,49,50,51,52]. Other cutaneous diseases related to worsening sexual function and reproduction have been lichen simplex [53], vitiligo, and chronic urticaria [54].

To our knowledge, this is the first review study to assess the prevalence of SD in AD. The strengths of this review are the inclusion of any article, regardless of language, the presentation of the methodology, and the report according to PRISMA guidelines, as well as the assessment of the risk of bias of the included studies. In terms of limitations, the great heterogeneity of the SD measurement instruments means that conclusions should be taken with caution. Furthermore, in a way that is not consistent with the studies included, it is very likely that there is a bias on the part of the informant due to the sensitivity of the subject matter. Finally, not being able to perform a meta-analysis of the data would have given a much more robust estimator of our conclusions.

## 5. Conclusions

In conclusion, we can establish that unlike other psychological comorbidities such as anxiety and depression, sexual dysfunction in AD is a field scarcely explored in the literature. This sexual dysfunction focuses on the male sex in large population studies and in clinical diagnoses without exploring it through specific and validated questionnaires in this regard. Further studies focused on both sexes are needed. It is important to correlate this sexual dysfunction with the severity of the disease, previous treatments, and cardiovascular comorbidities.

## Figures and Tables

**Figure 1 life-11-01314-f001:**
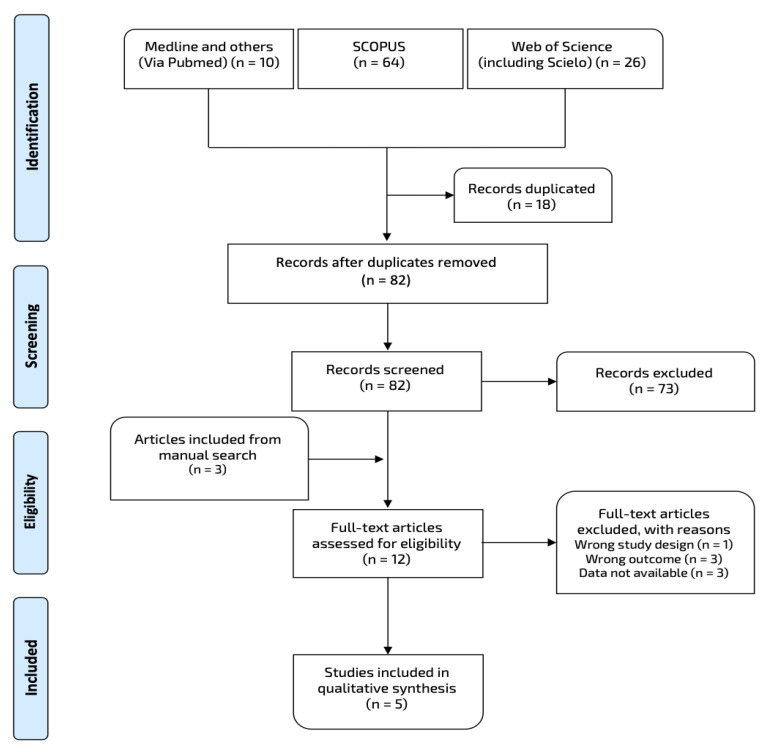
Flow chart of data extraction throughout the databases.

**Figure 2 life-11-01314-f002:**
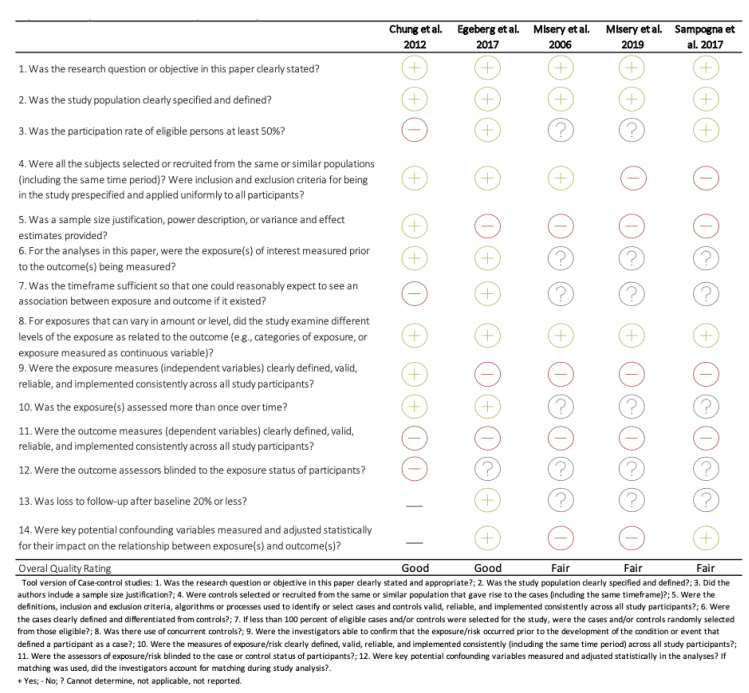
Quality assessment tool for observational cohort and cross-sectional studies.

**Table 1 life-11-01314-t001:** Main characteristics of studies regarding atopic dermatitis and sexual disfunction.

Reference	Country	Design	Gender/Age (x ± sd, Years)	Sample Size Cases (AD), N	Atopic Dermatitis Severity, N	Sample Size Controls, N	Sexual Dysfunction Assessment Tool	Prevalence SD in Cases, No (%)	Prevalence SD in Controls, No (%)
Chung et al., 2012 [18]	Taiwan	Case-Control	All Men/50.6 ± 15.0	3,997	Not reported	Healthy controls, 19,985	Erectile dysfunction (impotence, organic (International Classification of Diseases, 9th Revision, Clinical Modification (ICD-9-CM)	425 (10.6)	Healthy controls 1333 (6.7)
Egeberg et al., 2017 [17]	Denmark	Cohort Study	All men/53.0 ± 14.6, 46.7 ± 12.0, 56.3 ± 13.8, for general population, AD and psoriasis respectively	2,373	Mild AD, 1072 Severe AD, 1301	Healthy controls, 1,727,770 Psoriasis, 26,536	Erectile Dysfunction (patients’ first claimed prescription for drugs used in treatment of male ED (i.e., sildenafil, ATC code G04BE03; tadalafil, ATC code G04BE08; vardenafil, ATC code G04BE09; or avanafil, ATC code G04BE10).	112 (10.65) *	Healthy controls, 79,374 (10.79) * Psoriasis, 1533 (14.56) *
Misery et al., 2006 [24]	United Kingdom	Cross-sectional	91 males and 175 females/32.7 ± 12.7	266	Mild AD,4 Moderate AD, 114 Severe AD, 148	No controls	Sexual questionnaire from the CHOQ, which include 7 items for the patients and 6 for their partners. DLQI	153 (57.5)	-
Misery et al., 2019 [26]	France	Cross-sectional	427 males and 596 women	1022	Mild AD, 283 Moderate AD, 414 Severe AD, 327	No controls	Item nine from the DLQI	483 (57.9)	-
Sampogna et al., 2017 [25]	Multicenter Study from 13 European Countries; main author from Italy.	Cross-sectional	1524 males and 1950 females/38.3% <39 years; 34.2% from 40 to 59 years; 27.5% > 60 years.	448	Not reported	Psoriasis, 537 Acne, 202 Blistering disorders, 53 Other dermatological conditions ^+^, 2245	Item nine from the DLQI	130 (29.0)	Psoriasis, 187 (34.8) Blistering disorders, 18 (34.9)

Abbreviations: AD, Atopic Dermatitis; CHOQ, Cohorte HBP Observatoire et Qualité de Vie; DLQI, Dermatology Life Quality Index; SD, Sexual Dysfunction; sd, standard desviation. * Incidence of sexual dysfunction in atopic dermatitis patients. ^+^ Includes hidradenitis suppurativa, prurigo, urticaria, infection of the skin, pruritus, collagenosis, hair disorders, lichen planus, hyperhidrosis, ulcus cruris, seborrheic dermatitis, psychodermatological conditions, granuloma annulare, alopecia areata, vitiligo, rosacea, melanoma, benign skin tumors, nevi and non-melanoma skin cancer (individual data for each condition is not reported).

## Data Availability

Not applicable.

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
