# Peer review of "Sexual Dysfunction and Atopic Dermatitis: A Systematic Review"

_life, 2021, doi:10.3390/life11121314_

Round 1
Reviewer 1 Report
This manuscript aims to extensively analyze the relationship between sexual dysfunction and atopic dermatitis.
If we consider the focus of this paper, the description of the clinical features of atopic dermatitis in the introduction section is too broad, while it seems to me that in describing sexual dysfunction Authors mix clinical problems with physiological evolution of sexuality
There is also a lack of references on relevant statements, e.g., line 46: “around 10% of the world population...10-20% in children”…I’m not sure that it is so high…line 84-86 concerning prevalence of ED and sharing of risk factors with cardiovascular diseases ...
In overall terms, the significance of the article sounds poor and the only relevant meaning is to suggest that it is a topic that has been poorly studied and perhaps should be further investigated.
In addition, if the high prevalence of AD is in the pediatric -adolescent age, the analysis should have also been about the development of sexuality and gender relations in adolescents.
Author Response
Author reply:
Dear reviewer
We thank you for the revision made to our manuscript and we will comment on the reflections you make on it.
- a) The introduction may be a bit long, but we wanted to reflect the problems involved in the approach to sexual dysfunction as a psychological comorbidity in AD and the logical sequence that has led us to this research topic
- b) All statements have been carefully referenced, i.e. the prevalence percentages correspond to references 1 and 2, which are quite up-to-date.
- c) We do not consider that development is poor, as it is a topic little explored in the literature and for this reason, we have encouraged it to be the starting point of a PhD research. Furthermore, in order to guarantee the methodological quality of the work, it has been developed following the Preferred Reporting Items for Systematic Reviews and Meta-Analyses (PRISMA), and the methodological quality of the included studies has been measured using the Cochrane Risk of Bias Tool 2. Both aspects, as well as a methodology of peer review process and extraction of results, make, from our point of view, that the work has a methodological development of high scientific quality.
- d) As I mentioned in the previous point, this manuscript is the basis of a Ph D Research that has passed an approval process by the Ethics Committee of our hospital and the evaluation of adult patients with AD was chosen for its ethical implications. Obviously, there may be sexual dysfunction in adolescent patients, but it is not the objective of this work and, furthermore, in successive research studies, currently underway we are going to relate this psychological comorbidity with other physiological ones inherent to AD.
Reviewer 2 Report
Please, don´t use "believe" in a scientific manuscript, the correct approach in this instance is "in your experience or evaluation" - see page 5, line 214.
Author Response
Author reply:
Dear Reviewer
We thank you for the revision made to our manuscript and we will comment on the reflections you make on it.
We have proceeded to make the suggested change.
Regards
Reviewer 3 Report
A systematic review exploring the relationship between atopic dermatitis and sexual dysfunction; the article will be eligible for publication after revisions:
Although you correctly selected the studies to include in this review, you weren't able to highlight an important association between ad and sexual dysfunction, differently from other dermatological conditions....how do you explain this data?
in the introduction, it should be useful to divide AD according to clinical subtypes, and also to describe current treatments; here two articles you could embody:doi: 10.18176/jiaci.0519. and doi: 10.1111/dth.12787.
Author Response
Author reply:
Dear Reviewer
Thank you for the review of the paper. We procced to answer the ítems requested
The main explanation that we give that it has not been possible to demonstrate the existence of this association lies in three main points that we can summarize in the conclusions of our study:
- a) It is a subject scarcely explored in the published literature.
- b) Most studies do not use validated questionnaires.
- c) Most studies focus on males since they equate sexual dysfunction with erectile dysfunction.
Furthermore, despite not having found results as strong and conclusive as those found in other dermatological conditions, the studies analyzed seem to point to the impact of AD on patients' sex lives, which raises an incredibly extensive line of research from which to continue working.
Precisely for all these reasons, as stated in the introduction, it is comorbidity to be explored in our patients with AD.
Following the reviewer suggestion, we have expanded the introduction, adding the references indicated by the reviewer.
Regards
Reviewer 4 Report
An interesting systematic review about the prevalence of sexual dysfunction in AD; the argument is interesting and not fully addressed in the medical literature. The paper is very well structured, although the absence in the literature of work specifically addressing these problems, but only using questionnaires may render it hard to draw significant conclusions (and the principal topic was mainly erectile dysfunction). Only minor revisions:
page 2 line 45; a more wide description of the various subtypes of atopic dermatitis is in my opinion necessary; here is an interesting article: doi: 10.18176/jiaci.0519.
The fact that specific sexual dysfunctions associated with AD with the exception of erectile disfunction were never addressed by medical literature is a concept that must be strengthened.
Author Response
Author reply:
Dear Reviewer
We procced to answer the ítems requested
We greatly appreciate the comments made on our manuscript and we proceed to add the reference that you have indicated. We believe that the importance of your comment is now reinforced in the discussion.
Regards